# Difluoromethylation of (hetero)aryl chlorides with chlorodifluoromethane catalyzed by nickel

Chang Xu[1], Wen-Hao Guo[1], Xu He[1], Yin-Long Guo[1], Xue-Ying Zhang[1] & Xingang Zhang [1]

Relatively low reactivity hinders using chlorodifluoromethane (ClCF$_2$H) for general difluoromethylation with organic molecules, despite its availability as an inexpensive industrial chemical. To date, transformations of ClCF$_2$H are very limited and most of them involve difluorocarbene intermediate. Here, we describe a strategy for difluoromethylation of aromatics through nickel-catalyzed cross-coupling of ClCF$_2$H with readily accessible (hetero)aryl chlorides. The reaction proceeds under mild reaction conditions with high efficiency and features synthetic simplicity without preformation of arylmetals and broad substrate scope, including a variety of heteroaromatics and commercially available pharmaceuticals. The reliable practicability and scalability of the current nickel-catalyzed process has also been demonstrated by several 10-g scale reactions without loss of reaction efficiency. Preliminary mechanistic studies reveal that the reaction starts from the oxidative addition of aryl chlorides to Ni(0) and a difluoromethyl radical is involved in the reaction, providing a route for applications of ClCF$_2$H in organic synthesis and related chemistry.

[1] Key Laboratory of Organofluorine Chemistry, Center for Excellence in Molecular Synthesis, Shanghai Institute of Organic Chemistry, University of Chinese Academy of Sciences, Chinese Academy of Sciences, 345 Lingling Road, 200032 Shanghai, China. These authors contributed equally: Chang Xu, Wen-Hao Guo. Correspondence and requests for materials should be addressed to X.Z. (email: xgzhang@mail.sioc.ac.cn)

**D**ifluoromethylation of organic molecules using chlorodi-fluoromethane (ClCF$_2$H), an inexpensive industrial raw material used for production of fluorinated polymers[1], represents a cost-efficient and straightforward route to the synthesis of paramount important fluorinated compounds[2–12]. Its activation and transformation, however, is still of great challenge, due to the strong C–Cl bonding in this gaseous compound. Thus far, most transformation paths of ClCF$_2$H involve the difluor-ocarbene intermediate. The difluorocarbene species formed through pyrolysis at high temperature or through dehydro-chlorination under strong basic conditions has very limited synthetic applications: the former is only applied to produce tetrafluoroethylene (TFE) and related fluorinated polymers (e.g., Teflon)[1], and the latter is used to prepare heteroatom-substituted difluoromethylated compounds[13–16]. Very recently, we have developed a palladium-catalyzed difluoromethylation of arylbor-ons with ClCF$_2$H, also via a difluorocarbene pathway (Fig. 1a)[17]. This study has also demonstrated that the activation of ClCF$_2$H by transition metal can be conducted under mild reaction con-ditions and thus has wide application potential.

For both the practical application and the fundamental research, replacing the palladium catalyst by a first-row-based transition metal catalyst would pave a new and more cost-efficient way for applications of ClCF$_2$H in organic synthesis and medicinal chemistry. Instead of ClCF$_2$H activation, only rare examples of nickel catalyzed difluoroalkylation have been repor-ted for the cross-coupling of difluoroalkyl chloride (ClCF$_2$CO$_2$Et) with nucleophilic arylboronic acids, in which the C–Cl bond is activated by an ester group (CO$_2$Et) adjacent to the difluor-ocarbon[18–20]. As an inert substrate, the direct cleavage of C–Cl bond in ClCF$_2$H remains a great challenge. Herein, we report a nickel-catalyzed reductive cross-coupling between ClCF$_2$H and aryl chlorides[21–27], representing an alternative strategy for the fluoroalkylation reactions. The reaction proceeds under mild reaction conditions with high efficiency and enables difluor-omethylation of a variety of inexpensive and readily accessible aryl chlorides, including heteroaromatics and commercially available pharmaceuticals. The reaction can also extend to aryl bromides. In contrast to the difluorocarbene intermediate involved in the previous palladium-catalyzed process, the current Ni-catalyzed difluoromethylations undergoes a difluoromethyl radical pathway through the direct cleavage of C–Cl bond in ClCF$_2$H.

## Results

**Optimization of the Ni-catalyzed cross-coupling.** We began our studies on nickel-catalyzed reductive cross-coupling between ClCF$_2$H and aryl chlorides (Fig. 1b). The use of aryl chlorides is because of their cheapness, ready availability, and more impor-tantly, we proposed a direct transformation that can avoid the need for preformed arylmetals, such as arylborons and arylzincs. A suitable nickel catalytic system is the key to realize this synthesis route. To date, however, such a nickel catalyzed reductive cross-coupling between organohalides and fluor-oalkylated electrophiles has not been reported and remains a challenge, because of the difficulties in selectively controlling the catalytic cycle to suppress the side reactions, such as the forma-tion of hydrodehalogenated and dimerized fluorinated by-products. Although important progresses have been achieved in nickel-catalyzed reductive cross-coupling between aryl halides and unactivated alkyl halides[28,29], specific challenges still exist in similar reactions of inert aryl chlorides without substitution by electron-withdrawing groups[30], which is subject to the coupling reaction in the current study.

Accordingly, 1-(*tert*-butyl)-4-chlorobenzene **2a** was chosen as the model substrate for this nickel-catalyzed difluoromethylation reaction (Table 1). Without additives, either no reaction or low yields of difluoromethylated arene **3a** were observed in most of the experiments. The addition of MgCl$_2$[31] (1.5 equiv.) benefited the reaction with 16% yield of **3a** obtained when the reaction was carried out with NiCl$_2$·DME (10 mol%, DME, dimethoxyethane), bpy **L1** (10 mol%, bpy, 2,2′-bipyridine) and Zn (2.0 equiv.) in DMA (dimethylacetamide) at 80 °C (entry 1). But other additives, such as HCl, TMSCl (trimethyl chlorosilane), or DIBAL-H (diisobutyl aluminum hydride) showed no reactivities (Supple-mentary Table 13). Encouraged by this result, a survey of the reaction parameters, such as ligands, nickel sources, and solvents, was conducted. Unfortunately, no significant improvement of the reaction efficiency was observed (entries 2–4 and Supple-mentary Tables 1–2). Since a more electron-rich nickel center can benefit the oxidative addition to C–Cl bond, the combination of two electron rich ligands **L4** and DMAP (4-dimethylaminopyr-idine)[32,33] provided **3a** in 46% yield (entry 5). Other pyridine-based ligands were also examined, but were inferior to DMAP (entries 6–8). Decreasing the reaction temperature to 60 °C with 10 mol% of NiCl$_2$ could improve the yield of **3a** to 59% (entry 9). Further optimization of the reaction conditions

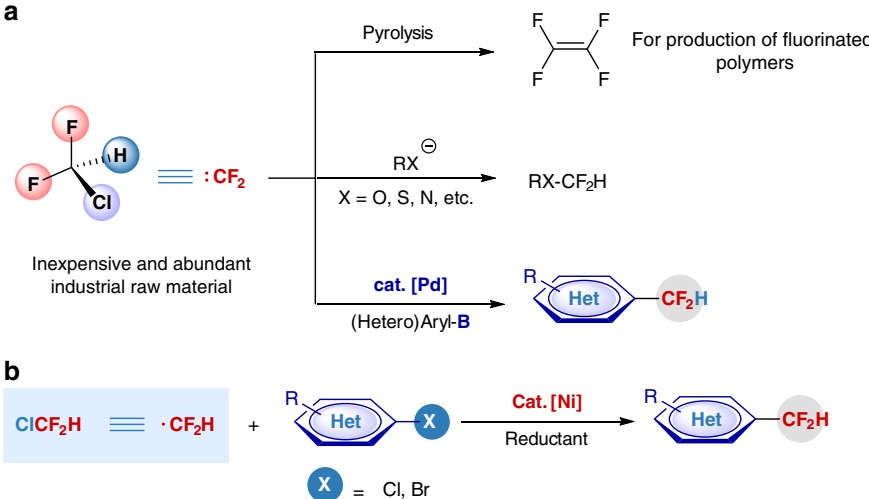

**Fig. 1** Strategies in activation of ClCF$_2$H. **a** Previous work, activation of ClCF$_2$H via a difluorocarbene pathway. **b** This work, a new activation of ClCF$_2$H through a difluoromethyl radical pathway

**Table 1 Representative results for the optimization of Ni-catalyzed difluoromethylation of 2a with ClCF$_2$H[a]**

| Entry | L (mol%) | Additive (x) | yield[b] (%), 3a |
|---|---|---|---|
| 1 | L1 | MgCl$_2$ (1.5) | 16 |
| 2 | L2 | MgCl$_2$ (1.5) | 21 |
| 3 | L3 | MgCl$_2$ (1.5) | 10 |
| 4 | L4 | MgCl$_2$ (1.5) | 33 |
| 5 | L4 + DMAP (20) | MgCl$_2$ (1.5) | 46 |
| 6 | L4 + Py (20) | MgCl$_2$ (1.5) | 35 |
| 7 | L4 + 4-MeOPy (20) | MgCl$_2$ (1.5) | 38 |
| 8 | L4 + 4-CF$_3$Py (20) | MgCl$_2$ (1.5) | 31 |
| 9[c] | L4 + DMAP (20) | MgCl$_2$ (4.0) | 59 |
| 10[d] | L4 + DMAP (20) | MgCl$_2$ (4.0) | 79 |
| 11[e] | L4 + DMAP (20) | MgCl$_2$ (4.0) | 77 |
| 12[f] | L4 + DMAP (20) | MgCl$_2$ (4.0) | NR |
| 13[g] | DMAP (20) | MgCl$_2$ (4.0) | NR |

*NR* no reaction
[a]Reaction conditions (unless otherwise specified): **1** (2.6 M in DMA, 6.5 equiv.), **2a** (0.2 mmol, 1.0 equiv.), DMA (2 mL)
[b]Determined by $^{19}$F NMR using fluorobenzene as an internal standard
[c]NiCl$_2$ (10 mol%) and Zn (3.0 equiv.) were used and reaction run at 60 °C
[d]NiCl$_2$ (15 mol %), **L4** (10 mol%), Zn (3.0 equiv.), and 3 Å MS were used and reaction run at 60 °C
[e]NiBr$_2$ (15 mol%), **L4** (10 mol%), Zn (3.0 equiv.), and 3 Å MS were used and reaction run at 60 °C
[f]Reaction run in the absence of nickel catalyst
[g]Reaction run in the absence of **L4**

(Supplementary Tables 4–12) revealed that the addition of 3 Å molecular sieves (MS) in conjunction with NiCl$_2$ (15 mol%) and **L4** (10 mol%) could afford **3a** in 79% yield (entry 10). In parallel, NiBr$_2$ as an alternative catalyst under the same reaction conditions could also lead to **3a** in a comparable yield (entry 11). It should be mentioned that the use of 10 mol% of NiCl$_2$ and 10 mol% of **L4** with 3 Å MS could also lead to **3a** in a comparable yield (76%) sometimes. But in most of the cases, we obtained the yields of **3a** in a range of 46% to 76%. We supposed that the use of excessive NiCl$_2$ vs **L4** was probably because a comproportionation occurred between [Ni$^{II}$] and in situ generated [Ni$^0$]. Switching Zn with organic reductant tetrakis(dimethylamino) ethylene (TDAE) led to no **3a** (Supplementary Table 14). The absence of nickel or **L4** failed to provide **3a** either (entries 12 and 13). Thus, these findings demonstrate that Ni/**L4** and Zn play an essential role in promotion of the reaction.

**Scope of the Ni-catalyzed cross-coupling.** With the viable reaction conditions in hand, a variety of aryl chlorides were examined (Table 2). For the electron-neutral aryl chlorides, the use of 10 mol% of NiCl$_2$ and **L4** in a low amount as 5 mol% still provided corresponding difluoromethylated arenes in high yields (**2b–2d**, **2x**, and **2y**). Although the *ortho* substituted substrates, such as *ortho* methyl, fluoride, vinyl and ester substituted phenyl chlorides furnished the difluoromethylated products in lower yields (**2e–2g** and **2n**), they are still synthetically useful for medicinal chemistry to access otherwise unavailable compounds. Aryl chlorides bearing electron-donating substituents were also amenable to the reaction, leading to **3h–3k** in moderate to good yields (**2h–2k**), in which methoxyl group at *meta* position provided higher yield than that at *para* position (**2i** and **2h**). Thus, the current nickel-catalyzed process

shows the much larger substrate scope than the previous nickel-catalyzed reductive cross-coupling, where the electron-rich and -neutral aryl chlorides have no reaction or only lead to poor yields[30]. Electron-deficient aryl chlorides were also competent coupling partners, highlighting the generality of this approach (**2l–2r**). The reaction exhibited good tolerance to functional groups including base or nucleophile-sensitive moieties, such as alkoxycarbonyl and enolizable ketone, and other groups such as vinyl, methylsulfonyl, nitrile, and substituted piperazine (**2g**, **2l–2t**). Remarkably, alcohol and arylboronate did not interfere with the reaction efficiency, and led to compounds **2u–2w** in good yields, featuring the advantages of this approach. Furthermore, the reaction can also extend to aryl bromides, both electron-rich and electron-deficient aryl bromides were suitable substrates (**2c′**, **2i′**, and **2l′**).

Synthesis of fluorinated heteroaromatic compounds is highly relevant to medicinal chemistry. To our delight, heteroaryl chlorides were also amenable to the current nickel-catalyzed reductive cross-coupling (Table 2). Pyridine-, quinoline-, and benzooxazole-containing substrates all underwent the reaction smoothly, leading to corresponding difluoromethylated heteroarenes in moderate to good yields (**4a-4i**).

We also examined this protocol for direct difluoromethylation of aryl chloride containing pharmaceuticals, since difluoromethyl group is considered as a bioisostere of hydroxyl and thio groups, and also as a lipophilic hydrogen bond donor[34,35] (Table 3). Commercially available drugs such as fenofibrate, clofibrate, chlorodiphenhydramine and sibutramine underwent the current nickel-catalyzed process smoothly and afforded corresponding difluoromethylated products in good yields (**6a-6d**). The good tolerance of trialkyl amines of this coupling provides a useful route for the modulation of biologically active molecules. *N*-Heterocycles containing drugs were also viable in the reaction.

**Table 2 Scope of the nickel-catalyzed reductive cross-coupling of ClCF₂H with aryl chlorides[a]**

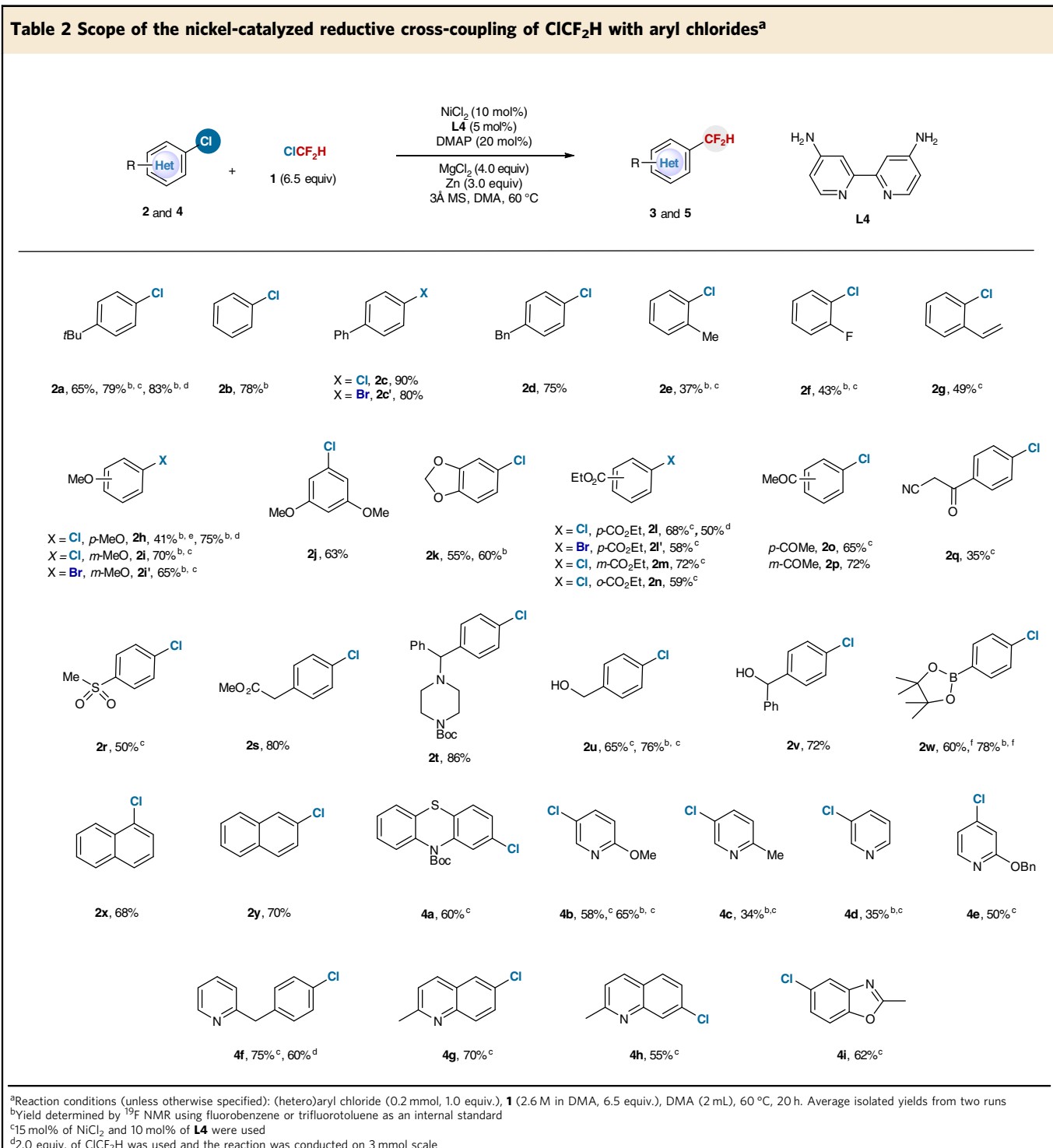

[a]Reaction conditions (unless otherwise specified): (hetero)aryl chloride (0.2 mmol, 1.0 equiv.), **1** (2.6 M in DMA, 6.5 equiv.), DMA (2 mL), 60 °C, 20 h. Average isolated yields from two runs
[b]Yield determined by ¹⁹F NMR using fluorobenzene or trifluorotoluene as an internal standard
[c]15 mol% of NiCl₂ and 10 mol% of **L4** were used
[d]2.0 equiv. of ClCF₂H was used and the reaction was conducted on 3 mmol scale
[e]20 mol% of NiBr₂ and 10 mol% of **L4** with or without 3 Å MS were used
[f]20 mol% of NiCl₂ and 10 mol% of **L4** were used

For instance, clomipramine and buclizine furnished the desired products in good yields (**6e** and **6f**). Although lorcaserin bearing a free amine afforded low yield, the protected *N*-Boc-lorcaserin led to difluoromethylated arene efficiently (**6g**). Furthermore, *N*-heteroaryl containing drug loratadine was also applicable to the cross-coupling and provided **7h** in 82% yield (**6h**). Most importantly, the acetyl protected empagliflozin, a drug used for the treatment of type II diabetes could also provide difluoromethylated product (**6i**). Although only 25% yield of **7i** was obtained, this strategy can trade off the yield for

fast synthesis of various interesting new biologically active molecules in the late stage without the need for multi-step parallel synthesis. The rebamipide derivative with unprotected amide bond was also a competent coupling partner (**6j**). This finding encouraged us to highlight the utility of this protocol further. As shown in **6k**, the direct difluoromethylation of protic groups containing drug tolvaptan without protection of hydroxyl and amide bond produced corresponding difluoromethylated product in a yield as high as 92%. Thus, this protocol provides a synthetic simplicity route for the

**Table 3 Late-stage difluoromethylation of pharmaceuticals[a]**

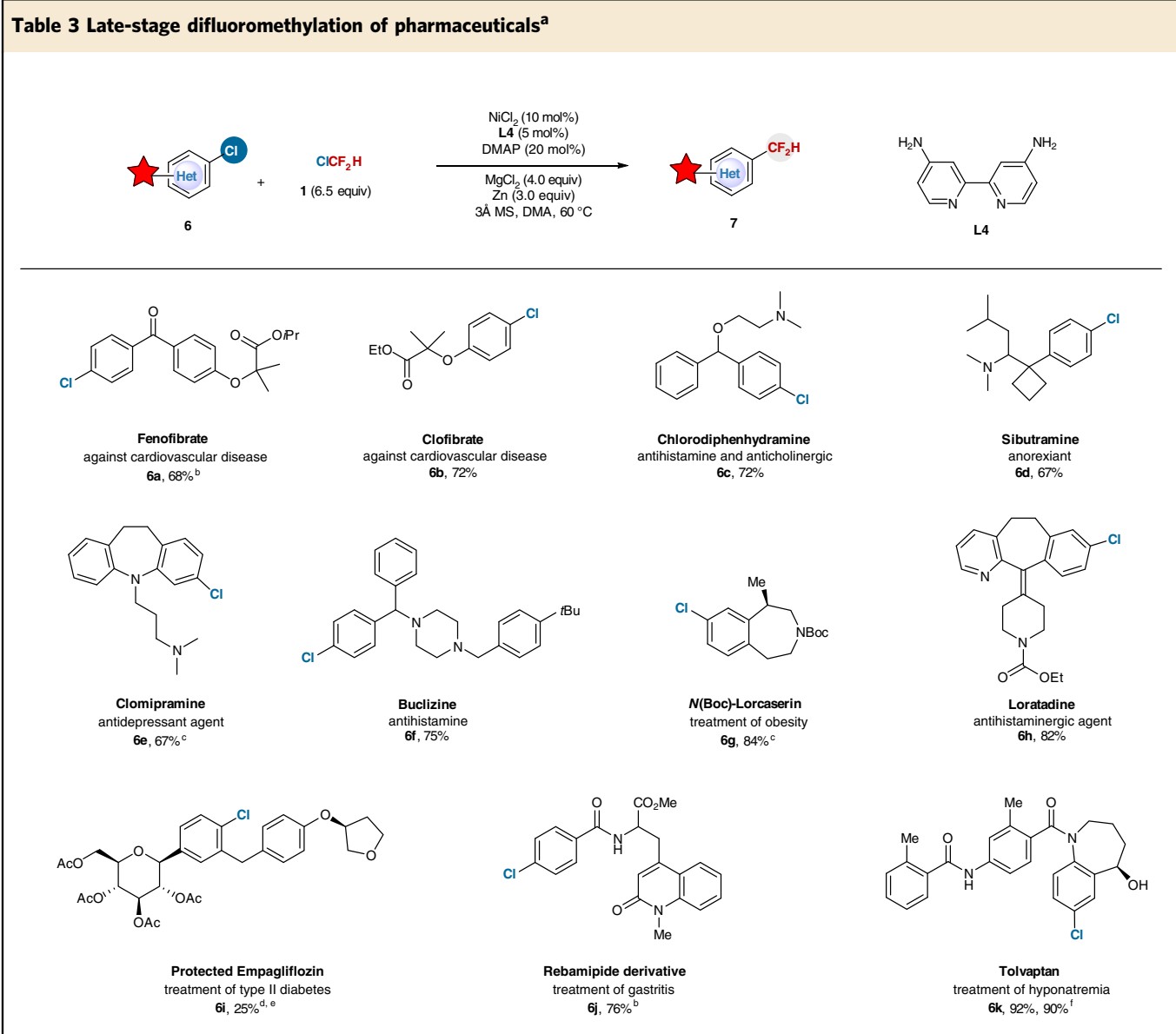

Reaction scheme:

6 + ClCF$_2$H (1, 6.5 equiv.) →
NiCl$_2$ (10 mol%), **L4** (5 mol%), DMAP (20 mol%)
MgCl$_2$ (4.0 equiv), Zn (3.0 equiv), 3Å MS, DMA, 60 °C
→ 7

**L4**

**Fenofibrate**
against cardiovascular disease
**6a**, 68%[b]

**Clofibrate**
against cardiovascular disease
**6b**, 72%

**Chlorodiphenhydramine**
antihistamine and anticholinergic
**6c**, 72%

**Sibutramine**
anorexiant
**6d**, 67%

**Clomipramine**
antidepressant agent
**6e**, 67%[c]

**Buclizine**
antihistamine
**6f**, 75%

**N(Boc)-Lorcaserin**
treatment of obesity
**6g**, 84%[c]

**Loratadine**
antihistaminergic agent
**6h**, 82%

**Protected Empagliflozin**
treatment of type II diabetes
**6i**, 25%[d, e]

**Rebamipide derivative**
treatment of gastritis
**6j**, 76%[b]

**Tolvaptan**
treatment of hyponatremia
**6k**, 92%, 90%[f]

[a]Reaction conditions (unless otherwise specified): (hetero)aryl chloride (0.2 mmol, 1.0 equiv.), **1** (2.6 M in DMA, 6.5 equiv.), DMA (2 mL), 60 °C, 20 h. Average isolated yields from two runs
[b]20 mol% of NiBr$_2$ and 10 mol% of **L4** with or without 3 Å MS were used
[c]15 mol% of NiCl$_2$ and 10 mol% of **L4** were used
[d]Yield determined by $^{19}$F NMR using fluorobenzene as an internal standard
[e]20 mol% of NiCl$_2$ and 10 mol% of **L4** were used
[f]2.0 equiv. of ClCF$_2$H was used and the reaction was conducted on 3 mmol scale

applications in drug discovery and development. Most remarkably, decreasing the loading amount of ClCF$_2$H to 2 equiv. still provided difluoromethylated arenes with high efficiency as demonstrated by the synthesis of compounds **2a**, **2h**, **2l**, **4f** (Table 2), and **6k** (Table 3), thus demonstrating the advantages of this approach.

To demonstrate the scalability of the current nickel-catalyzed process, several 10-g scale reactions of aryl chlorides were conducted. As shown in Fig. 2a, reaction of ClCF$_2$H with 11.3 g of 4-chloro-1,1′-biphenyl **2c** proceeded smoothly under standard reaction conditions, providing **3c** in 80% yield. The electron-deficient aryl chloride **2l** (11.1g) was also applicable to the reaction and afforded **3l** even in a higher yield (74%) (Fig. 2b). Notably, substrate bearing a hydroxyl group (**2v**) could also furnish its corresponding difluoromethylated product **3v** in a

much higher yield (90%) (Fig. 2c). Most remarkably, even 10-g scale late stage difluoromethylation of pharmaceutical tolvaptan **6k**, a high yield (91%) was still obtained (Fig. 2d). It is also worthy to note that decreasing the loading amount of ClCF$_2$H to 2 equiv. could also lead to difluoromethylated arene without loss of reaction efficiency as shown by 10-g scale reaction of **2v** (Fig. 2c), thus demonstrating the good scalability and reliability of this reaction. In light of the wide existence of aryl chloride structural motif in pharmaceuticals and biologically active molecules, this approach would be useful in medicinal chemistry.

## Discussion

In the mechanism study of this reaction, we conducted several experiments. Firstly, to rule out the possible generation of

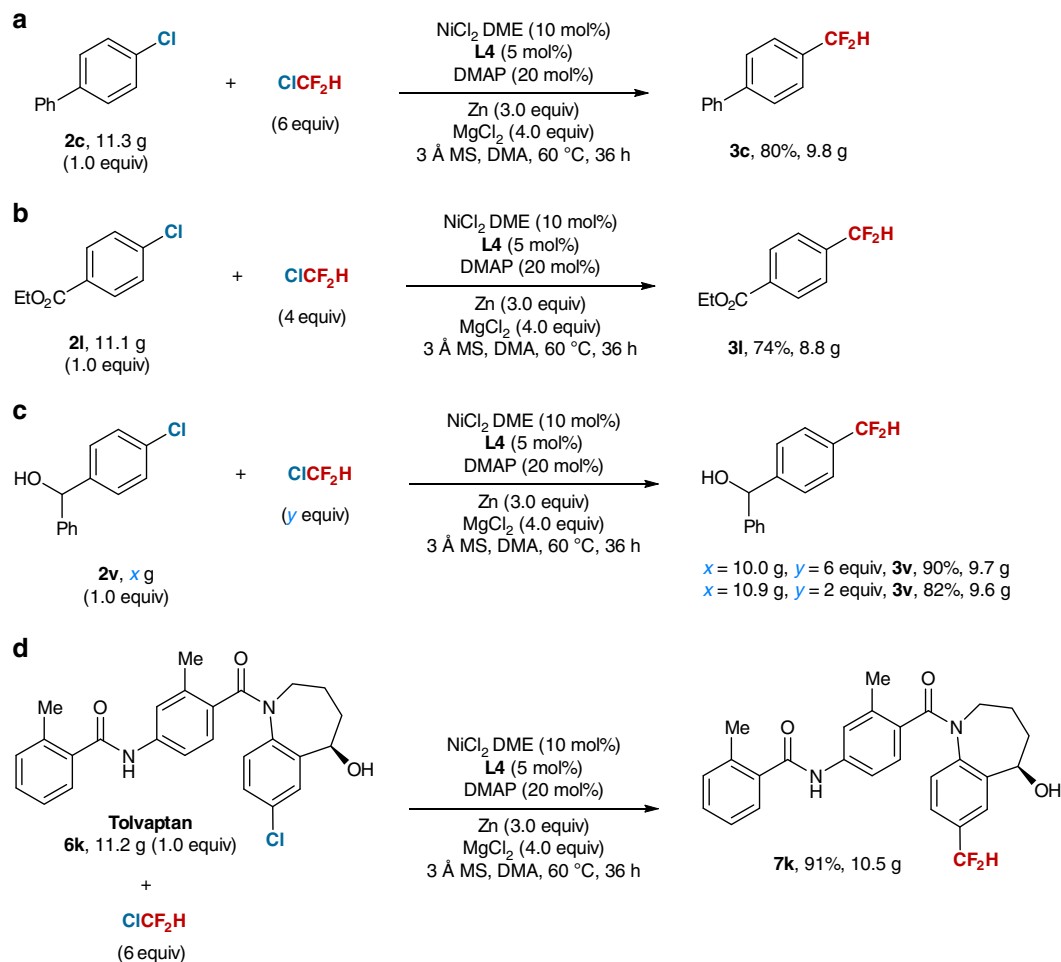

**Fig. 2** Ten-gram scale reaction of aryl chlorides with ClCF$_2$H. **a** Reaction of ClCF$_2$H with **2c**. **b** Reaction of ClCF$_2$H with **2l**. **c** Reaction of ClCF$_2$H with **2v**. **d** Reaction of ClCF$_2$H with **6k**

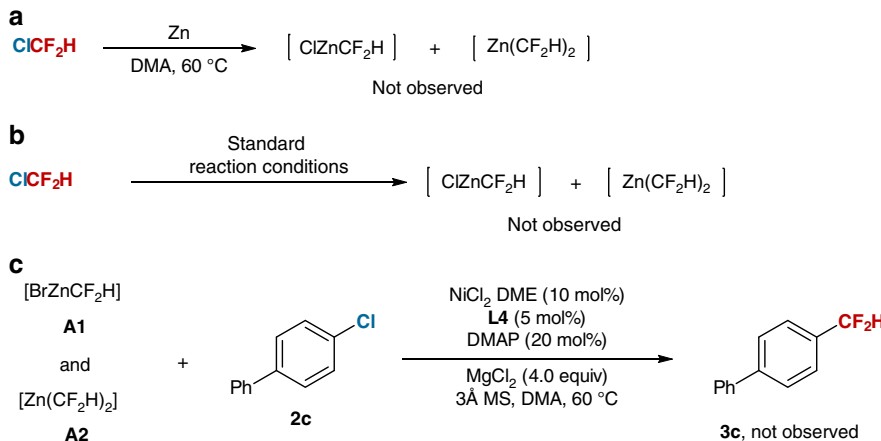

**Fig. 3** Reaction of ClCF$_2$H with zinc. **a** Reaction of ClCF$_2$H with zinc in DMA. **b** Reaction of ClCF$_2$H with zinc under standard reaction conditions. **c** Reaction of arylchloride **2c** with difluoromethyl zinc species

difluoromethyl zinc species[10,11] in situ between ClCF$_2$H and Zn, we performed control experiments. No difluoromethyl zinc species were observed in the reaction of ClCF$_2$H with Zn in DMA at 60 °C, or even treatment of ClCF$_2$H under standard reaction conditions in the absence of aryl chlorides (Fig. 3a, b). Instead, only starting material ClCF$_2$H was observed after the reaction. We also prepared difluoromethyl zinc species (**A1** and

**A2**) by reaction of BrCF$_2$H with Zn in DMA at 60 °C (Supplementary Methods)[36]; however, no desired product **3c** was obtained when these difluoromethyl zinc species were treated with aryl chloride **2c** under standard reaction conditions (Fig. 3c). Thus, these results exclude the pathway that the formation of difluoromethylated arenes is derived from the cross-coupling between difluoromethyl zinc species and aryl

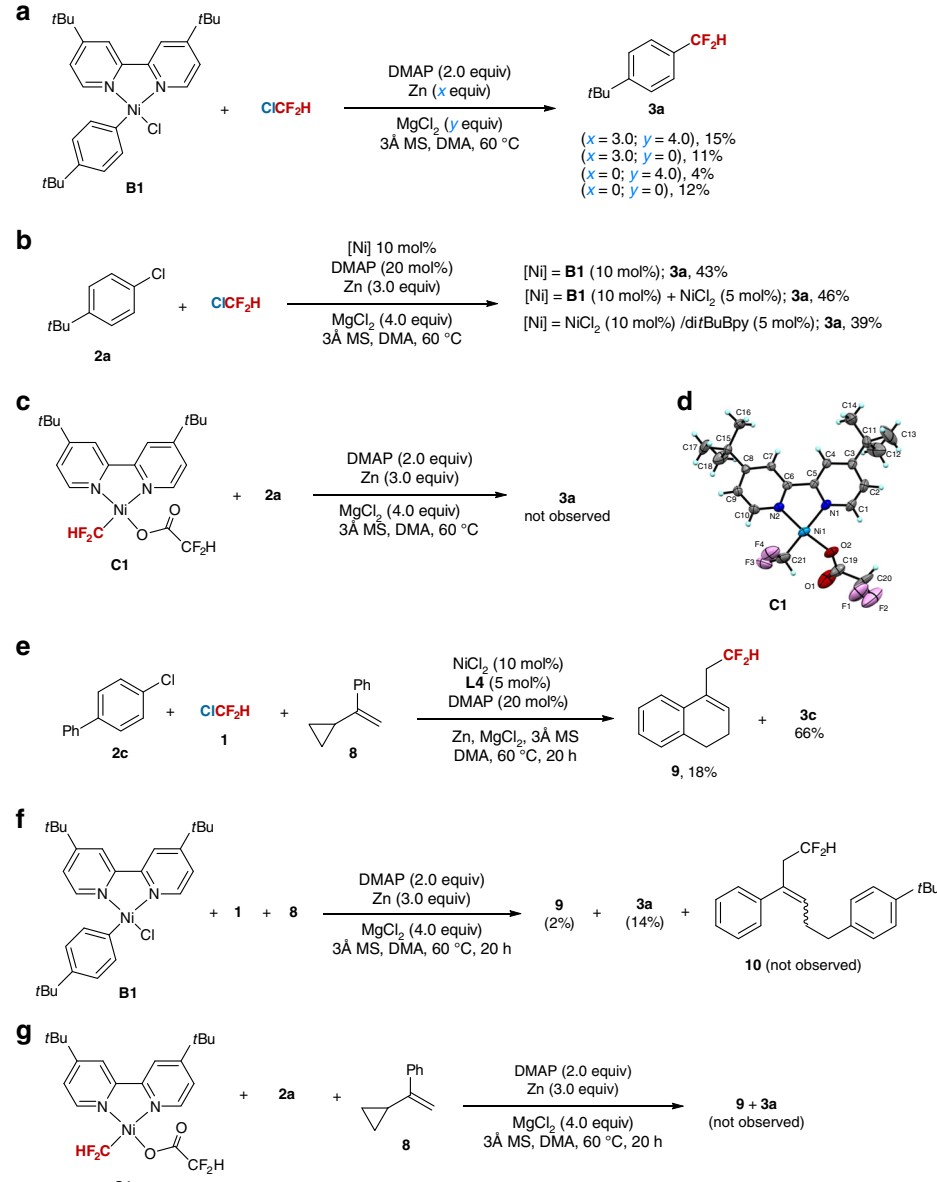

**Fig. 4** Mechanistic studies. **a** Reaction of nickel complex **B1** with ClCF₂H. **b** **B1** or NiCl₂/di*t*BuBpy catalyzed reaction of **2a** with ClCF₂H. **c** Reaction of **C1** with **2a**. **d** X-ray crystal structure of **C1**. **e** Experiments to trap the difluoromethyl radical by reaction of **2c** and **8** with ClCF₂H. **f** Reaction of **B1** and **8** with ClCF₂H. **g** Reaction of **2a** and **8** with **C1**

chlorides. On the basis of the previous reports[28,29], we suggest that a nickel-based, reductive cross-coupling catalytic cycle is involved in the reaction.

Secondly, to identify the initiation of current reaction from aryl chloride or ClCF₂H, we prepared aryl nickel complex [*p*-*t*Bu-PhNi(di*t*BuBpy)Cl] (**B1**)[37] and difluoromethyl nickel complex [HCF₂Ni(di*t*BuBpy)HCF₂CO₂] (**C1**)[38,39]. The structure of **C1** was confirmed by X-ray crystallographic analysis (Fig. 4d). To the best of our knowledge, the preparation of difluoromethyl nickel(II) complex has not been reported so far. The use of 4,4′-di*t*Bu-Bpy (**L2**) instead of 4,4′-diNH₂-Bpy (**L4**) is because of the difficulties in isolation of [Ar-Ni(**L4**)-Cl] and [CF₂H-Ni(**L4**)-HCF₂CO₂]. In addition, **L2** could also promote the reaction under standard reaction conditions and provided **3a** in 39% yield (Fig. 4b). A 15% yield of **3a** was provided when **B1** was treated with ClCF₂H under standard reaction conditions (Fig. 4a). Complex **B1** could also serve as a precatalyst and provided **3a** in 43% yield, which is comparable with the yield

obtained by using NiCl₂/di*t*BuBpy catalytic system (Fig. 4b). However, no **3a** was obtained by reaction of difluoromethyl nickel complex **C1** with aryl chloride **2a** (Fig. 4c), demonstrating that the current reaction is initiated from aryl chloride and the possibility that the reaction starts from the oxidative addition of ClCF₂H to Ni(0) is unlikely.

We also performed control experiments to gain some mechanistic insights into the reaction further (Fig. 4a). The omission of Zn led to **3a** in only 4% yield, implying that an active, low-valent nickel species is needed to promote the catalytic cycle by reduction of Ar-Ni^II (**B1**) with Zn. Inspired by the previous report, in which a reduction of Ar-Ni^II to Ar-Ni^I by reducing metals was proposed in the nickel-catalyzed coupling of aryl chlorides[40], we envisioned that similar pathway may be involved in the reaction. Furthermore, a lower yield (11%) of **3a** was provided without MgCl₂ (Fig. 4a), indicating that the presence of MgCl₂ in current nickel-catalyzed process is probably to facilitate the reduction of nickel(II) complex by Zn to generate active

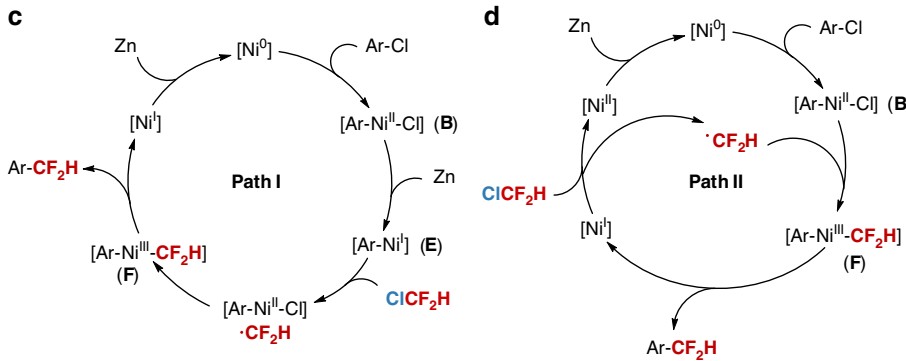

**Fig. 5** The role of DMAP. **a** [NiCl$_2$(di$t$BuBpy)] (**D1**) catalyzed reaction between **2a** and **1** with or without DMAP. **b** [NiCl$_2$(DMAP)$_4$] (**D2**) catalyzed reaction between **2a** and **1** with or without di$t$BuBpy. **c** Reaction of **B1** with ClCF$_2$H in the presence of DMAP. **d** Reaction of **B1** with ClCF$_2$H without DMAP

**Fig. 6** Proposed reaction mechanism. **a** Proposed mechanism via a radical-cage-rebound process. **b** Proposed mechanism via a radical chain process

nickel species. However, the exact role of MgCl$_2$ remains elusive. On the other hand, the omission of Zn/MgCl$_2$ could provide **3a** in 12% yield (Fig. 4a), suggesting that intermediate **B1** reduction is not needed without MgCl$_2$.

Thirdly, to probe whether a difluoromethyl radical existed in the reaction, several radical trapping experiments were conducted. Radical inhibition experiments showed that the reaction could be readily inhibited by addition of electron transfer scavenger 1,4-dinitrobenzene[22] or a radical scavenger 2,2,6,6-tetramethyl-1-piperidinyloxy (TEMPO) (Supplementary Table 16). In addition, radical clock experiment showed that a ring-expanded product **9** was formed in 18% yield when ClCF$_2$H was treated with α-cyclopropylstyrene **8** in the presence of aryl chloride **2c** under standard reaction conditions (Fig. 4e). However, when a radical scavenger TEMPO was added to the reaction, the reaction was totally inhibited without observation of compounds **9** and **3c** (Supplementary Methods). Compound **9** could also be obtained

by a stoichiometric reaction of nickel complex **B1** with ClCF$_2$H and **8** under standard reaction conditions (Fig. 4f). But difluoromethyl nickel complex **C1** failed to provide compound **9** (Fig. 4g), thus ruling out the possible formation of compound **9** from **C1** through the Ni-concerted insertion mechanism (Supplementary Fig. 142b). Furthermore, the possibility of formation of compound **9** from aryldifluoromethyl nickel complex Ar-[Ni]-CF$_2$H generated in situ between **B1** and ClCF$_2$H is also unlikely, as no ring-opening product **10** was observed by treatment of **B1** with ClCF$_2$H and **8** (Fig. 4f and Supplementary Fig. 142c). Therefore, these results suggest that the formation of **9** via a radical pathway (Supplementary Fig. 142a) is reasonable and a difluoromethyl radical species is involved in current catalytic cycle.

Finally, to establish the role of DMAP in the reaction, we prepared nickel complexes [NiCl$_2$(di$t$BuBpy)] (**D1**) and [NiCl$_2$(DMAP)$_4$] (**D2**)[27]. Both of them could serve as precatalysts

and provided **3a** in comparable yields (Fig. 5a, b). However, **D1** provided **3a** in a lower yield (16%) without DMAP (Fig. 5a) and no **3a** was observed by using **D2** in the absence of di*t*BuBpy (Fig. 5b). Additionally, a DMAP coordinated nickel complex **B2** was observed by reaction of **B1** with DMAP, which could also produce **3a** in a 15% yield (Fig. 5c). But only 6% yield of **3a** was obtained without DMAP (Fig. 5d). These results demonstrate that DMAP can function as a co-ligand to coordinate to the nickel center and thus facilitate the catalytic cycle.

We also performed a Hammett-type analysis of the reaction (Supplementary Fig. 137 and 138). Plots of $\log(k_{rel})$ versus $\sigma$ and $\sigma$ $(-)$ were linear with moderate quality ($R^2 \sim 0.93$), and the slope ($\rho$) was between 1.9 and 2.0[30,41,42]. The $\rho$ values are smaller than those reports on stoichiometric studies of the oxidative addition of aryl halides to Ni (4.4–8.8)[43,44], indicating that the oxidative addition of aryl chlorides to Ni(0) is not rate-determining step[45]. On the basis of these results and previous reports[28,29], we propose that the reaction starts from aryl chloride via a radical-cage-rebound process[29] (Fig. 6a). An oxidative addition of aryl chloride to Ni(0) initiates the reaction. Subsequently, the resulting nickel(II) complex [Ar-Ni$^{II}$-Cl] (**B**) is reduced by Zn to generate [Ar-Ni$^{I}$] (**E**). **E** undergoes the second oxidative addition with ClCF$_2$H to produce [Ar-Ni$^{III}$-CF$_2$H] (**F**) through a cage rebound process, in which a difluoromethyl radical ·CF$_2$H is produced via a single-electron-transfer pathway, subsequently, the resulting ·CF$_2$H rapidly recombines with [Ar-Ni$^{II}$-Cl] to give **F**. Finally, **F** undergoes reductive elimination to deliver difluoromethylated arenes and [Ni$^{I}$]. [Ni$^{I}$] is further reduced by Zn to regenerate [Ni$^{0}$]. Alternatively, a radical chain mechanism[28,46] is also possible (Fig. 6b) as the intermediate **B1** could also lead to difluoromethylated product without reduction by Zn (Fig. 4a). In this pathway, the ·CF$_2$H generated by reaction of [Ni$^{I}$] with ClCF$_2$H diffuses to the solution to combine with [Ar-Ni$^{II}$-Cl] **B** to produce **F**, which undergoes reductive elimination to give difluoromethylated product. Finally, the resulting [Ni$^{II}$] is reduced by Zn to regenerate [Ni$^{0}$] (Supplementary Fig. 144).

In conclusion, we have developed a practical nickel-catalyzed difluoromethylation of (hetero)aryl chlorides and bromides with abundant and inexpensive ClCF$_2$H, representing a new strategy for fluoroalkylation reactions. The reaction proceeds under mild reaction conditions and can efficiently access a wide range of difluoromethylated (hetero)aromatics, including pharmaceuticals. Comparing to the previous difluoromethylation methods[3–12,17], the current nickel-catalyzed process features several advantages, inexpensive ClCF$_2$H and low-cost nickel catalyst; more accessible and cheaper aryl chlorides as well as no need for preformed arylmetals; broad substrate scope including a variety of heteroaromatics and pharmaceuticals; synthetic simplicity and convenience without prefunctionalization of drugs and biologically active molecules. Particularly, the ability of direct modulation of pharmaceuticals by using ClCF$_2$H provides good opportunities to discover new medicinal agents. The additives MgCl$_2$ and DMAP are critical to the reaction efficiency and DMAP can serve as a co-ligand to facilitate the catalytic cycle. Preliminary mechanistic studies reveal that the reaction starts from the oxidative addition of aryl halides to Ni(0) and a difluoromethyl radical is involved in the reaction, which is in contrast to the previous difluorocarbene pathway, thus paving a new way for applications of ClCF$_2$H in organic synthesis and related chemistry.

## Methods

**General procedure for the nickel catalyzed cross-coupling**. To a 25 mL of Schlenk tube were added aryl chloride **2**, **4**, or **6** (0.2 mmol, 1.0 equiv.), NiCl$_2$ (10 mol%), **L4** (5 mol%), zinc dust (3.0 equiv.), MgCl$_2$ (4.0 equiv.), 3 Å MS (100 mg) and DMAP (20 mol%). The mixture was evacuated and backfilled with argon for

three times, DMA (2 mL) and ClCF$_2$H **1** (2.6 M in DMA, 1.3 mmol, 6.5 equiv.) were then added. The Schlenk tube was screw capped and put into a preheated oil bath (60 °C). After stirring for 20 h, the reaction mixture was cooled to room temperature and diluted with ethyl acetate (2 mL). The yield was determined by $^{19}$F NMR using fluorobenzene as an internal standard before working up. Then the reaction mixture was filtered with a pad of cellite. The filtrate was washed with brine, extracted with EtOAc for three times. Then the organic layer was dried over Na$_2$SO$_4$ and concentrated. The residue was purified with silica gel chromatography to give product **3**, **5**, or **7**. Isolated yield is based on the average of two runs under identical conditions.

**Data availability**. The authors declare that all the data supporting the findings of this study are available within the paper and its supplementary information files. CCDC 1572871 contains the supplementary crystallographic data for **C1**. These data can be obtained free of charge from the Cambridge Crystallographic Data Centre via www.ccdc.cam.ac.uk/data_request/cif.

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

## Acknowledgements

Financial support for this work was provided by the National Natural Science Foundation of China (21425208, 21672238, 21421002, and 21332010), the National Basic Research Program of China (973 Program) (No. 2015CB931900), the Strategic Priority Research Program of the Chinese Academy of Sciences (No. XDB20000000), and SIOC.

## Author contributions

C.X. and W.-H.G. contributed equally to this work. X.Z., C.X., and W.-H.G. conceived and designed the experiments. C.X. and W.-H.G. performed the experiments. C.X. conducted all the mechanism studies. X.H. and X.-Y.Z. prepared some starting materials. Y.-L.G. conducted MS analysis of nickel complexes. C.X. and W.-H.G. analyzed the data. X.Z. wrote the paper. All authors discussed the results and commented on the manuscript.

## Additional information

**Competing interests:** The authors declare no competing interests.

