## [Peer Review File(PDF 641 kb) · Nature Communications]

Reviewer #1 (Remarks to the Author):

This manuscript by Zhang and coworkers describes a nickel-catalyzed difluoromethylation of (hetero)aryl chlorides with chlorodifluoromethane. As an inexpensive industrial raw material, the reductive cross-coupling of ClCF₂H with aryl chlorides (or bromides) represents a cost-efficient and straightforward route for synthesis of difluoromethylated arenes. Compared with previous report on palladium-catalyzed process involving difluorocarbene intermediate by the same group, this transformation processed via a totally different radical pathway by direct cleavage of C-Cl bond in ClCF₂H. This method demonstrated also mild reaction conditions and broad substrate scope, and worked well for late-stage difluoromethylation of pharmaceuticals. Mechanistic studies revealed also that the reaction initiated from the oxidative addition of aryl halides to Ni(0) and a difluoromethyl radical is involved in the catalytic cycle, which offered a new way for applications of ClCF₂H in organic synthesis and related chemistry. Based on all advantages mentioned above, I recommend this manuscript to be published on Nature Communication but after some revisions shown below has been addressed:

1. As for the optimized catalyst system, the ratio of nickel and ligands (Ni/ditBuBpy/DMAP = 10/5/20) looks interesting, and changing the ratio of Ni/ditBuBpy/DMAP to 10/10/20 gave obviously lower yield. While the author proposed ditBuBpy/Ni/Ln was the active catalyst, how to explain why 5 mol% more of Nickel not coordinating to ditBuBpy was required in this catalytic system? Meanwhile, did Ni/DMAPx complex work in the catalytic cycle?
2. In all cases for mechanistic studies, B1 or NiCl₂(ditBuBpy) was added as the sole catalyst. While (Ni/ditBuBpy/DMAP = 10/5/20) was optimized as the best ration, some more NiCl₂ should be added into the reaction system to meet the optimized conditions.
3. As shown in the supplementary materials, the examination of ratio between nickel and ligands looks a little strange. Reaction conditions in entry 1, Supplementary Table 9 is the same as entry 1, Supplementary Tables 13, 14 and 15, but giving different yields.
4. As a member of Freon family, it'd better control the amount of ClCF₂H used in the reaction. 6.5 equiv of ClCF₂H is really excessive, maybe the authors can reinvestigate the conditions to decrease the amount of this Freon reagent.
5. It looks that ortho-substituents were not well compatible with this reaction. Could any other ortho-substituents, such as 2-Ph, ester or halogen, been tolerated in this system?
6. While difluoromethylation happened on the aryl ring of 4a, 4c-4f, only 4b are tolerated as a real heteroaryl substrate. How about the other heteroaryl chlorides?
7. Please comment the real role of 3 Å MS?
8. While ditBuBpy/Ni/Ln was proposed as the catalyst, please explain why the subjection of stoichiometric amount of B1 into standard conditions gave much lower yield than the catalytic reaction?

Reviewer #2 (Remarks to the Author):

The manuscript by Zhang and co-workers reports a nickel-catalyzed difluoromethylation of aryl and heteroaryl chlorides with ClCHF_2 . The reaction conditions were carefully optimized, and the reaction mechanism was investigated. My questions about this manuscript are as follows: (1) ClCHF_2 is an ozone-depleting substance (Freon-22), so this starting material is already prohibited from selling in Europe and USA (Montreal Protocol). This situation will prevent this synthetic methodology from wide application. (2) Although nickel is catalytic in the reaction, the amount of Zn is used in 3 equivalents. And surprisingly, MgCl_2 is used in 4 equivalents. Such kind of "recipe" makes me think that this reaction may not be easily scaled up. In the paper, the scale of reactions are only in 0.2 mmol (for aryl chlorides). The authors should demonstrate a couple of examples of large scale of the reactions (such as ten grams of aryl chlorides). (3) Aryl bromides are less efficient than chlorides in reactions. Why? (4) After a long discussion, the authors are still not sure whether the reaction proceeds through 6c or 6d, which is very disappointing to me. (5) As the authors correctly mentioned in the Introduction part, the nickel catalyzed difluoroalkylation has been reported for the cross-coupling of ClCF_2R (R is not H), so the concept of this paper is not really new. Overall, although the manuscript is of interest to some specialists, the results reported in the manuscript are not qualified for publication in Nature Communication.

Reviewer 1:

R1: As for the optimized catalyst system, the ratio of nickel and ligands (Ni/ditBuBpy/DMAP = 10/5/20) looks interesting, and changing the ratio of Ni/ditBuBpy/DMAP to 10/10/20 gave obviously lower yield. While the author proposed ditBubpy/Ni/Ln was the active catalyst, how to explain why 5 mol% more of Nickel not coordinating to ditBuBpy was required in this catalytic system? Meanwhile, did Ni/DMAPx complex work in the catalytic cycle?

Response to R1: In fact, the use of 10 mol% of NiCl₂ and 10 mol% of 4,4'-diNH₂-Bpy (**L4**) could also lead to **3a** in a comparable yield (76%) sometimes. However, in most of the cases, we obtained different yields under this condition. We supposed that the use of excessive NiCl₂ (5 mol% more of Ni) vs **L4** was probably because a comproportionation occurred between [Ni^{II}] and *in situ* generated [Ni⁰]. The use of Ni/DMAPx complex failed to provide the desired product, see Table 1 entry 13 and Fig. 5b, demonstrating that **L4** is essential for the reaction.

R2: In all cases for mechanistic studies, **B1** or NiCl₂(ditBuBpy) was added as the sole catalyst. While (Ni/ditBuBpy/DMAP = 10/5/20) was optimized as the best ration, some more NiCl₂ should be added into the reaction system to meet the optimized conditions.

Response to R2: It should be mentioned that the optimized reaction conditions are Ni/diNH₂Bpy (**L4**)/DMAP = 10/5/20. The use of 4,4'-ditBu-Bpy (**L2**) instead of 4,4'-diNH₂-Bpy (**L4**) is because of the difficulties in isolation of [Ar-Ni(**L4**)-Cl] and [CF₂H-Ni(**L4**)-HCF₂CO₂]. In addition, **L2** could also promote the reaction under standard reaction conditions and provided **3a** in 39% yield (Fig. 4b). The use of excessive NiCl₂ to meet the optimized conditions was conducted, providing **3a** in 35% to 46% yields (Fig 4b). The reason for the use of excessive NiCl₂ (5 mol% more of Ni) vs **L4** under the standard reaction conditions was probably because a comproportionation occurred between [Ni^{II}] and *in situ* generated [Ni⁰].

R3: As shown in the supplementary materials, the examination of ratio between nickel and ligands looks a little strange. Reaction conditions in entry 1, Supplementary Table 9 is the same as entry 1, Supplementary Tables 13, 14 and 15, but giving different yields.

Response to R3: The yields in Tables 13, 14 and 15 have been corrected. See revised Supplementary materials.

R4: As a member of Freon family, it'd better control the amount of ClCF_2H used in the reaction. 6.5 equiv of ClCF_2H is really excessive, maybe the authors can reinvestigate the conditions to decrease the amount of this Freon reagent.

Response to R4: The loading amount of ClCF_2H can be decreased to 2.0 equiv without loss of reaction efficiency (Fig 2c). Even a higher yield (82% vs 72% small scale, Table 2, 2v) was obtained when the reaction was scaled up to 10 g (Fig 2c).

R5: It looks that ortho-substituents were not well compatible with this reaction. Could any other ortho-substituents, such as 2-Ph, ester or halogen, been tolerated in this system?

Response to R5: Other *ortho*-substituted aryl chlorides, such as *ortho* fluoride, vinyl and ester substituted phenyl chlorides, were all suitable substrates, providing corresponding products in 43% to 59% yields (Table 2, **2f**, **2g** and **2n**). But 2-Ph phenyl chloride was not suitable substrate due to the steric effect.

Response to R6: While difluoromethylation happened on the aryl ring of 4a, 4c-4f, only 4b are tolerated as a real heteroaryl substrate. How about the other heteroaryl chlorides?

Response to R6: Other heteroaryl chlorides, such as 5-chloro-2-methylpyridine, 3-chloropyridine and 2-(benzyloxy)-4-chloropyridine, are all suitable substrates, providing corresponding difluoromethylated heteroarenes in synthetically useful yields (Table 2, **4c-4e**).

R7: Please comment the real role of 3Å MS?

Response to R7: The role of 3Å MS remains elusive at this stage, probably it can disperse zinc powder and stabilize the active nickel species.

R8: While ditBubpy/Ni/Ln was proposed as the catalyst, please explain why the subjection of stoichiometric amount of **B1** into standard conditions gave much lower yield than the catalytic reaction?

Response to R8: The low yield of **3a** is because of the instability of **B1**, which is prone to decomposition in solution.

Reviewer 2:

R1: CICHF₂ is an ozone-depleting substance (Freon-22), so this starting material is already prohibited from selling in Europe and USA (Montreal Protocol). This situation will prevent this synthetic methodology from wide application.

Response to R1: Although CClF₂H is an ozone-depleting substance (Freon-22) and is already prohibited from selling in Europe and USA (Montreal Protocol), it is an inexpensive (1 \$/kg) and abundant raw industrial chemical used for making fluorinated polymers. Every year million tons of CClF₂H is produced and we still can easily access it from industrial companies. Notably, many of commercially available difluoromethylating reagents are prepared from CClF₂H or XCF₂H. For example, the common difluoromethylating reagent TMSCF₂H can be prepared from CClF₂H (*J. Org. Chem.* **2003**, 68, 4457), but TMSCF₂H is 157730 to 62700 times expensive than CClF₂H (TMSCF₂H 157.73 \$/g, Aldrich; 62.7 \$/g, TCI vs CClF₂H 1 \$/kg). Therefore, from the point of view of cost efficiency and step economy, *the use of CClF₂H as a difluoromethylating reagent represents a most straightforward and cost-efficient approach to access difluoromethylated compounds.*

R2: Although nickel is catalytic in the reaction, the amount of Zn is used in 3 equivalents. And surprisingly, MgCl₂ is used in 4 equivalents. Such kind of "recipe" make me think that this reaction may not be easily scaled up. In the paper, the scale of reactions are only in 0.2 mmol (for aryl chlorides). The authors should demonstrate a couple of examples of large scale of the reactions (such as ten grams of aryl chlorides).

Response to R2: To demonstrate the scalability of current nickel-catalyzed process, five 10-gram scale reactions of aryl chlorides were conducted. As shown in Fig. 2a, reaction of CClF₂H with 11.3 g of 4-chloro-1,1'-biphenyl **2c** proceeded smoothly under standard reaction conditions, providing **3c** in 80% yield. The electron-deficient aryl chloride **2l** (11.1 g) was also applicable to the reaction and afforded **3l** even in a higher yield (74%) (Fig. 2b). Notably, substrate bearing a hydroxyl group (**2v**) could also furnish its corresponding difluoromethylated product **3v** in a much higher yield (90%) (Fig. 2c). Most remarkably, even 10-g scale late stage difluoromethylation of pharmaceutical **6k**, a high yield (91%) was still obtained (Fig. 2d). It is also worthy to note that decreasing the loading amount of CClF₂H to 2 equiv could also lead to difluoromethylated arene without loss of reaction efficiency as shown by 10-g scale reaction of **2v** (Fig. 2c), thus demonstrating the good scalability and reliability of this reaction.

R3: Aryl bromides are less efficient than chlorides in reactions. Why?

Response to R3: The slightly less reaction efficiency of aryl bromides compared with aryl chlorides is due to the formation of hydrodebrominated and dimerized arenes. However, the yields of aryl bromides (58% to 80%) were still moderate to good.

R4: After a long discussion, the authors are still not sure whether the reaction proceeds through 6c or 6d, which is very disappointing to me.

Response to R4: On the basis of our mechanistic studies and previous reports (ref 28 and 29), we proposed two possible pathways, radical-cage-rebound pathway (**Path I**) and radical chain process (**Path II**), which might make reviewer 2 disappointed. However, previous computational mechanistic studies demonstrated that both of pathways (**Path I** and **Path II**) proposed in the manuscript are possible for the nickel-catalyzed reductive cross-couplings due to the small energy difference between these two possible pathways (ref 46 and *J. Organomet. Chem.* **2014**, 770, 130). Therefore, both of **Path I** and **Path II** are possible pathways for the current reaction, we cannot rule out one of them. Furthermore, our mechanistic studies reveal that the reaction starts from the oxidation of Ni(0) to aryl chlorides and a novel difluoromethyl radical is involved in the reaction, which offers a new way for applications of CICF_2H in organic synthesis and related chemistry. We believe that the current reaction will not only provide a useful instrument for drug discovery and development, but also prompt research in nickel catalyzed fluoroalkylation reactions with inexpensive fluoroalkyl halides.

R5: As the authors correctly mentioned in the Introduction part, the nickel catalyzed difluoroalkylation has been reported for the cross-coupling of CICF_2R (R is not H), so the concept of this paper is not really new.

Response to R5: Although we mentioned only rare examples (actually, two examples) of nickel catalyzed difluoroalkylation reactions have been reported for the cross-coupling of difluoroalkyl chlorides in the Introduction part (ref 18 and 19), they were nickel catalyzed cross-coupling of *$\text{CICF}_2\text{CO}_2\text{Et}$ with nucleophilic arylboronic acids*, in which the C-Cl bond is *activated by an ester group adjacent to the difluorocarbon*. As an inert substrate, the direct cleavage of C-Cl bond in CICF_2H is more difficult than that of $\text{CICF}_2\text{CO}_2\text{Et}$ due to the absence of activating group CO_2Et in CICF_2H . *The direct cleavage of C-Cl bond in CICF_2H remains a big challenge, and has not been reported.* However, *in this manuscript, we describe a totally new* strategy for the transition-metal-catalyzed fluoroalkylation from difluoroalkyl halide, namely, nickel-catalyzed *reductive cross-coupling of CICF_2H with electrophilic aryl chlorides*. This concept is totally different from previous work

(introduction part).

Furthermore, it is worthy to note that to date, the nickel catalyzed reductive cross-coupling between aryl chlorides and nonfluorinated alkyl halides remains a challenge, because electron-rich and electron-neutral aryl chlorides were not suitable substrates. While, with the cheapest fluorine source ClCF_2H as difluoromethylating reagent, the current nickel catalyzed process enable difluoromethylation of a variety of aryl chlorides, including a wide range of pharmaceuticals. Since many natural products and pharmaceuticals contain aryl chloride structural motif, the ability of this method to directly introduce a difluoromethyl group into biologically active molecules provides good opportunities to use ClCF_2H for the synthesis and development new medicinal agents. Compared to the previous activation of ClCF_2H via a difluorocarbene pathway, *the current reaction through a difluoromethyl radical pathway offered a new way for applications of ClCF_2H in organic synthesis and related chemistry*. Therefore, the concept described in this manuscript is totally new in terms of the polarization and activation of the inert CHF_2Cl molecule, as well as the Ni-catalyzed cross-coupling.

Reviewer #1 (Remarks to the Author):

As this reviewer mentioned in the first round of review, this nickel-catalyzed difluoromethylation of aryl chlorides and ClCF_2H is really important and interesting. The abundant and inexpensive ClCF_2H , an ozone-depleting gas (Freon-22), could be well used in this new transformation as a difluoromethylating source. While many questions has been raised in the first round of review, this revised manuscript gives good replies to almost every question that both reviewers concerned. Accordingly, I definitely support its publication on Nature communications, but after the following minor revisions:

1) As shown in the revised manuscript and coverletter, "In fact, the use of 10 mol% of NiCl_2 and 10 mol% of 4,4'-diNH₂-Bpy (L4) could also lead to 3a in a comparable yield (76%) sometimes." Indeed, this yield is even better than the result obtained with 10 mol% of NiCl_2 and 5 mol% of 4,4'-diNH₂-Bpy (L4) used (65% yield, 2a, Table 2), and the best result (79% yield, entry 10, Table 1) was afforded using up to 15% nickel catalyst. So, this reviewer can not understand "However, in most of the cases, we obtained different yields under this condition."

Actually, the above mentioned result should be added into Table 1, and some more substrates in Table 2 should be performed under these conditions. These results will be very helpful for further optimization of reaction conditions.

As we can see from the supplementary materials, the optimization of reaction conditions is really not reasonable enough.

2) This reviewer still concern about the amount of ClCF_2H (Freon-22). While the author indicated that a special example (2v) could afford even higher yield using 2.0 equiv of ClCF_2H when the reaction was scaled up to 10 g, how about the model substrate in Table 1 and some common substrates in Table 2?

3) When the authors answered the questions from Reviewer 2 in the coverletter, the corresponding viewpoints should be added into the manuscript to show the advantages of this method, especially for R1 and R5.

Reviewer 1:

R1: As shown in the revised manuscript and cover letter, “In fact, the use of 10 mol% of NiCl₂ and 10 mol% of 4,4'-diNH₂-Bpy (**L4**) could also lead to **3a** in a comparable yield (76%) sometimes.” Indeed, this yield is even better than the result obtained with 10 mol% of NiCl₂ and 5 mol% of 4,4'-diNH₂-Bpy (**L4**) used (65% yield, **2a**, Table 2), and the best result (79% yield, entry 10, Table 1) was afforded using up to 15% nickel catalyst. So, this reviewer can not understand “However, in most of the cases, we obtained different yields under this condition.” Actually, the above mentioned result should be added into Table 1, and some more substrates in Table 2 should be performed under these conditions. These results will be very helpful for further optimization of reaction conditions.

As we can see from the supplementary materials, the optimization of reaction conditions is really not reasonable enough.

Response to R1: In fact, the use of 10 mol% of NiCl₂ and 10 mol% of 4,4'-diNH₂-Bpy (**L4**) could also lead to **3a** in a comparable yield (76% determined by ¹⁹F NMR) sometimes. But its repeatability was poor, in most of the cases, we obtained the yields of **3a** in a range of 46% to 76% (determined by ¹⁹F NMR). We have added these comments in the text (highlighted by yellow), but to avoid misleading the readers, we didn't add this data into the Table 1. For the 65% yield of **3a** in Table 2, it is an isolated yield. While the 79% yield of **3a** shown in Table 1 is the yield determined by ¹⁹F NMR. It should be mentioned that all of the yields shown in Table 1 were determined by ¹⁹F NMR, however, in Table 2, except volatile products, all the yields were isolated yields. That is the reason we have two yields for **2a** (79% determined by ¹⁹F NMR, 65% isolated yield). The lower isolated yield of **3a** is because of its volatility, which results in loss of the product during the purification process.

We also used 10 mol% of NiCl₂ and 10 mol% of 4,4'-diNH₂-Bpy (**L4**) as the reaction conditions to prepare other products in Table 2. However, we did not obtain positive results. Aryl chlorides bearing electron-withdrawing groups showed no activity under these reaction conditions (**2l**, 3%; **6a**, nr). Electron-rich and electron-neutral substrates led to comparable or lower yields than that obtained with standard reaction conditions (**2h**, 54%; **2u**, 69%; **2v**, 34%; **6k**, 24%, determined by ¹⁹F NMR). The heteroaryl chlorides also provided corresponding products in lower yields (**4b**, 51%; **4h**, 9% determined by ¹⁹F NMR). Thus, these results demonstrate that the standard reaction conditions are the optimal reaction conditions, which not only provide the difluoromethylated arenes with repeatable and higher yields, but also enable the difluoromethylation of aryl chlorides with broad substrate scope.

R2: This reviewer still concern about the amount of ClCF_2H (Freon-22). While the author indicated that a special example (**2v**) could afford even higher yield using 2.0 equiv of ClCF_2H when the reaction was scaled up to 10 g, how about the model substrate in Table 1 and some common substrates in Table 2?

Response to R2: When the loading amount of ClCF_2H was decreased to 2 equiv, the model substrate **2a** led to corresponding product in a comparable yield (83% determined by ^{19}F NMR, Table 2 **2a**). Other substrates including **2h**, **2l**, **4f** and **6k** were also examined under these reaction conditions and provided difluoromethylated arenes with high efficiency. We have added these data in the Table 2.

R3: When the authors answered the questions from Reviewer 2 in the cover letter, the corresponding viewpoints should be added into the manuscript to show the advantages of this method, especially for R1 and R5.

Response to R3: We have added the corresponding viewpoints in the text.

Reviewer #1 (Remarks to the Author):

Zhang and coworkers did really nice job during the revision of this manuscript, and all questions we concerned have been well answered right now. As a result, this reviewer here give a clear support for its publication on Nature Communicatians.